# The Relationship between Political, Economic, Social, and Cultural Vulnerability and Food Insecurity among Adults Aged 50 Years and Older

**DOI:** 10.3390/nu13113896

**Published:** 2021-10-29

**Authors:** Patrick J. Brady, Natoshia M. Askelson, Sato Ashida, Faryle Nothwehr, Brandi Janssen, David Frisvold

**Affiliations:** 1Division of Epidemiology and Community Health, School of Public Health, University of Minnesota, 420 Delaware Street SE, Minneapolis, MN 55455, USA; 2Department of Community and Behavioral Health, College of Public Health, University of Iowa, 145 N Riverside Drive, Iowa City, IA 52246, USA; natoshia-askelson@uiowa.edu (N.M.A.); sato-ashida@uiowa.edu (S.A.); Faryle-nothwehr@uiowa.edu (F.N.); 3Health Policy Research Program, Public Policy Center, University of Iowa, 310 S Grand Ave., Iowa City, IA 52242, USA; 4Department of Occupational and Environmental Health, College of Public Health, University of Iowa, 145 N Riverside Drive, Iowa City, IA 52246, USA; brandi-janssen@uiowa.edu; 5Department of Economics, Tippie College of Business, University of Iowa, 21 E Market Street, Iowa City, IA 52242, USA; david-frisvold@uiowa.edu; 6Social and Education Policy Research Program, Public Policy Center, University of Iowa, 310 S Grand Ave., Iowa City, IA 52242, USA

**Keywords:** social exclusion, food insecurity, older adults

## Abstract

Individuals experience food insecurity when they worry about or have limited access to nutritious foods. Food insecurity negatively impacts older adults’ health. Social exclusion is a theoretical framework describing how unequal access to rights, resources, and capabilities results in political, economic, social, and cultural vulnerability, which leads to health disparities. We used the Health and Retirement Study to cross-sectionally examine associations between vulnerability and experiencing food insecurity in adults 50 years and older using the social exclusion framework. We tested the association between experiencing food insecurity and indicators of political, economic, social, and cultural vulnerability using logistic regression controlling for demographic and health-related factors. Analyses were performed with all respondents and sub-group of respondents with incomes less than 400% of the federal poverty level (FPL). Assets (OR = 0.97 in both samples), income (OR = 0.85, 0.80 in 400% FPL sub-sample), perceived positive social support from other family (OR = 0.86, 0.84 in 400% FPL sub-sample), and perceived everyday discrimination (OR = 1.68, 1.82 in 400% FPL sub-sample) were significantly associated with food insecurity. Perceived positive social support from spouses, children, or friends and U.S. citizenship status were not significantly associated with food insecurity. Further research is needed to define and measure each dimension of vulnerability in the social exclusion framework. Interventions and policies designed to prevent food insecurity should address these vulnerabilities.

## 1. Introduction

Food insecurity is a substantial public health problem affecting approximately one in ten adults aged 50 years and older [1,2]. The prevalence of food insecurity in the United States in 2020 was 10.5% [3], which is a significant decrease from the prevalence of 14.3% in 2013 when the data analyzed in this manuscript was collected [4]. Despite the fact that prevalence of food insecurity has been decreasing, and that food insecurity is less prevalent in older adults, for example with 6.9% [3] and 8.7% [4] of households with older adults experiencing food insecurity in 2020 and 2013, respectively, addressing food insecurity in older adults should remain public health priority as it negatively impacts their health and well-being. Food insecurity is associated with poor dietary intake [5,6,7,8,9] negative physical and mental health outcomes [8,9,10,11,12], and overall chronic disease burden [13]. Food insecure older adults are more likely to report needing more assistance with activities of daily living [7], impaired functional status [14], and lower quality of life [15] when compared to food-secure older adults. Food insecurity among older adults leads to higher health care utilization [16,17] and costs [18,19]. Reducing or preventing food insecurity will have many individual and societal benefits. In order to effectively meet the food needs of older adults, it is important to have a theoretical understanding of factors associated with increased or decreased risk of experiencing food insecurity.

Social exclusion is a theoretical framework describing how different mechanisms that reduce an individual’s access to rights, resources, and capabilities leads to vulnerability along four dimensions (i.e., political, economic, social, and cultural) that interact to produce observed health outcomes [20]. For a more complete description of the social exclusion framework, see Adam & Potvin’s 2016 manuscript [20], but we have adapted their framework into a simplified conceptual model for this study in Figure 1. Social exclusion has been previously used as a theoretical framework to understand social disadvantage, health, and well-being among older adults [21,22,23], but to our knowledge it has not been used to examine food insecurity in this population.

Many of the risk factors for food insecurity conceptually align with the four dimensions of vulnerability. Markers of political status, such as citizenship or immigration status, determine eligibility for programs such as SNAP and other government safety-net programs, such as Medicaid. Studies have shown that food assistance programs such as SNAP and home-delivered meals programs are effective at reducing the risk of experiencing food insecurity [24,25], and older adults who are ineligible for SNAP due to their immigrant status are at increased risk of experiencing food insecurity [26]. There is also evidence showing that having adequate health insurance may impact food insecurity. For example, Medicaid expansion under the Affordable Care Act is associated with a reduction in rates of very low food security among low-income, nonelderly childless adults, whom the program targets [27]. Economic factors such as socioeconomic status and renting rather than owning a home have been shown to be associated with experiencing food insecurity [1,3]. Food insecurity can be impacted by any situation that limits an individual’s financial resources [28], such as utility costs [29] and medical expenditures [30,31]. Social networks and connections have been shown to be related to food acquisition strategies in under-resourced areas [32] and are used as coping strategies among food insecure households [33,34]. Additionally, lack of perceived social support has been shown to be associated with food insecurity [35,36], and social support has been shown to be a major factor in food exchanges among food insecure older adults [37]. Finally, culturally stigmatized groups such as Black or Hispanic households are more likely to experience food insecurity [1,3].

Researchers should empirically test the relationships proposed by theoretical frameworks such as social exclusion in order to confirm that the proposed framework has a basis in reality [38,39], but there is a gap in the literature around how political, economic, social, and cultural vulnerability are related to experiencing food insecurity. The Health and Retirement Study (HRS) is a nationally representative survey of adults aged 50 years and older in the United States and offers an opportunity to fill this gap. The HRS collects vast amounts of information on health and wellness, socioeconomic factors, and psychosocial factors through multiple data collections and has been extensively described elsewhere [40,41,42,43].

The aim of this study is to demonstrate the relationship between vulnerability as defined by the social exclusion framework and food insecurity among adults aged 50 years and greater by conducting a cross-sectional, secondary analysis of HRS data. We hypothesized that each of the following measures of vulnerability will be significantly negatively associated with the likelihood of experiencing food insecurity among adults aged 50 years and older: Hypothesis 1 assets and income (economic vulnerability); Hypothesis 2 perceived positive social support from spouse/partner, children, other family, and friends and living arrangement (social vulnerability); and Hypothesis 3 U.S. citizenship status (political vulnerability); and that the following measures of vulnerability will be significantly positively associated with the likelihood of experiencing food insecurity among adults aged 50 years and older: Hypothesis 4) perceived everyday discrimination (cultural vulnerability). If the relationships hypothesized between different dimensions of vulnerability and food insecurity are supported by the HRS data, the social exclusion framework can be used to develop and inform interventions, programs, and policies aimed at reducing or preventing food insecurity.

## 2. Materials and Methods

### 2.1. Study Population and Sample

The study population for this analysis is community-dwelling adults aged 50 years and older in the United States. Publicly available datasets were analyzed in this study. The data used in this study can be found at https://hrs.isr.umich.edu/about (accessed on 4 October 2021). The sample consists of households with individuals aged 50 years and older from the HRS who had at least one respondent complete both the 2013 Health Care and Nutrition Survey and the 2012 Leave Behind Questionnaire, were not living in a nursing home, and did not have a proxy complete any of the data collections. Figure 2 shows the process for obtaining the analytic sample for this study.

The food insecurity measure used in this study is intended to be at the household level, but some households had multiple respondents complete the Health Care and Nutrition Survey. To reduce the relatedness of responses in the sample, we only included one respondent per household. To select which respondent to retain in the analytic sample, we compared the food security status of each individual in the same household. In cases where one respondent identified the household as food secure and one identified the household as food insecure, we retained the respondent who identified as food insecure as their household is likely facing some amount of food insufficiency. In households where multiple respondents identified the same household food security status, one individual was randomly selected.

Because HRS imputation documentation recommends addressing outliers for imputed variables [44], we removed cases with imputed values three standard deviations above or below the mean value. This resulted in 2460 cases in the analytic sample for the regression analysis. An additional dataset was created including only individuals with incomes less than 400% of the Federal Poverty Level (FPL) in 2012 (USD 60,520), so that we could conduct a sub-sample analysis. There were 1725 cases in the analytic sample for the less than 400% FPL regression analysis.

### 2.2. Measures

The main outcome for this study is food security status. The Health Care and Nutrition Survey uses the 6-item USDA short-form food security questions to establish household food security, modified for mailed surveys [45]. Responses to each of the five items were scored according to USDA methods [45] and the resulting value was dichotomized as food insecure (summed score = 2 to 6) or food secure (summed score = 0 or 1).

We identified potential indicators for each dimension of vulnerability. For economic vulnerability, we used financial and housing assets and yearly income because they reflect major sources of monetary resources, financial equity, and wealth available to the household. Each of these measures was a continuous variable with each one-unit increase representing an additional USD 10,000 in income or assets. For social vulnerability, we used measures of perceived positive social support from spouses, children, other family, and friends and whether the respondent was living alone. The Leave Behind Questionnaire has three items (How much do they really understand the way you feel about things? How much can you rely upon them if you have a serious problem? And how much can you open up to them if you need to talk about your worries?) to assess perceived positive social support from each of these groups [46,47]. Possible values for each of the social vulnerability items ranged from 0 to 3. In 2012, each of the scales had an alpha reliability score between 0.80 and 0.87 among participants who completed the Leave Behind Questionnaire [46]. For individuals who indicate they do not have a spouse, children, other family, or friends and skipped the question due to the questionnaire’s skip pattern, we set the value for their responses to the lowest value (“Not at all”) because they were not receiving support from that source. Whether the respondent was living alone was entered as a binary variable (Living alone vs. not living alone). For cultural vulnerability, we used a 6-item version of the Everyday Discrimination Scale based on a 5-item scale with an additional item added for the context of older adults [46,48,49]. The six items were: in your day-to-day life, how often have any of the following things happened to you: (1) You are treated with less courtesy or respect than other people, (2) You receive poorer service than other people at restaurants or stores, (3) People act as if they think you are not smart, (4) People act as if they are afraid of you, (5) You are threatened or harassed, and (6) You receive poorer service or treatment than other people from doctors or hospitals. Possible values ranged from 0 to 5. In 2012, the 6-item scale had an alpha reliability score of 0.83 among participants who completed the Leave Behind Questionnaire [45]. We used U.S. citizenship status as the indicator for political vulnerability. This measure was based on a single item asking if individuals not born in the United States were citizens. We combined the indicator variable for assets by summing their values and for social and cultural vulnerability by averaging their values and creating interval measures.

We also included control variables for factors previously shown to be significantly associated with food insecurity or which were theoretically relevant. We controlled for demographic factors (race, Hispanic ethnicity, age, gender, education level, and marital status [3]), perceived and actual health-related factors (self-rated health [7,8,14], limitations in activities of daily living [9], Center for Epidemiologic Studies Depression scale score [8,9,10], and number of chronic health conditions [13]), and two factors related to healthcare costs (having health insurance [27] and total medical expenditures [31,32]). Race (White vs. non-White), ethnicity (Hispanic vs. non-Hispanic), gender (Males vs. Female), marital status (married/partnered vs. separated/divorced/widowed/never married), and health insurance (has vs. does not have) were coded as binary variables. Education was entered as a four-level categorical variable (less than high school, high school or equivalent, some college, and college degree or above). Age, self-rated health, limitations in activities of daily living, CESD scale-score, number of chronic health conditions, and medical expenditures were continuous variables. The medical expenditure variable was constructed similar to the assets and income variables, with each one-unit increase representing an additional USD 10,000 in expenditures.

### 2.3. Analysis

We produced descriptive statistics for all variables. We then ran a logistic regression to examine the association between food security status with indicators for each of the dimensions of vulnerability while controlling for receiving SNAP benefits in the previous year, race, Hispanic ethnicity, age, gender, marital status, education level, self-rated health, having health insurance, medical expenditures, limitations in activities of daily living, CESD scale score, and number of chronic health conditions. All variables were entered according to their type (binary, ordinal, or continuous), with the reference groups identified where appropriate. The model was estimated using full information maximum likelihood estimation methods with robust standard errors. This same procedure was repeated using the sub-sample of respondents with incomes less than 400% of the FPL, and our results are reported for each sample. Analyses were conducted using Stata version 15 and MPLUS version 8.4.

## 3. Results

### 3.1. Descriptive Statistics

Descriptive statistics for categorical (Table 1) and continuous (Table 2) variables used in the regression analysis are presented below. About 26% of respondents with any income experienced food insecurity, while about 33% of respondents in the sub-sample with incomes less than 400% FPL experienced food insecurity. Respondents in the samples were mostly white (72% and 69%), Non-Hispanic (88% and 58%), identified as female (63% and 67%), and were not SNAP participants (87% and 83%). For the political vulnerability indicator, about 12% of the sample with any incomes and slightly over 13.5% of the sample with incomes less than 400% FPL were not U.S. citizens. For the social vulnerability indicators, about 28% of the sample with any incomes and 34% of the sample with incomes less than 400% FPL were living alone and the greatest amount of perceived positive social support came from children and the lowest from spouses in both income groups. The perceived everyday discrimination measure did not meaningfully differ between respondents with all incomes and with incomes less than 400% FPL. For respondents with any income, the average amount of assets was USD 181,500, and the average yearly income was USD 51,500. For respondents with incomes less than 400% FPL, the average amount of assets was USD 115,400, and the average yearly income was USD 26,500.

### 3.2. Logistic Regression

The results of the logistic regression analysis are shown in Table 3. The results for indicators of vulnerability did not meaningfully differ between the two samples. When significant, each indicator of vulnerability behaved in the expected direction. For each one-unit increase in perceived positive social support from other family members, the odds of experiencing food insecurity were reduced by 14% and 16% for individuals with any incomes and individuals with incomes less than 400% FPL. All other sources of perceived positive social support and living alone were not significantly associated with experiencing food insecurity with odds ratios ranging from 0.96 to 1.16. For each one-unit increase on the perceived everyday discrimination scale, the odds of experiencing food insecurity were increased by 68% and 82% for individuals with any income and individuals with incomes less than 400% FPL, respectively. For every additional USD 10,000 in assets, there was a 3% reduction in the odds of experiencing food insecurity in both samples. For every additional USD 10,000 per year in income, there were 15% and 20% reductions in the odds of experiencing food insecurity for individuals with any income and individuals with incomes less than 400% FPL, respectively. U.S. citizenship status was not significantly associated with food insecurity in either model.

## 4. Discussion

This study demonstrated the relationship between vulnerability as defined in the social exclusion framework and the risk of experiencing food insecurity in adults aged 50 years and older. Hypotheses 1 and 4 were supported, with both economic vulnerability measures and the cultural vulnerability measure significantly associated with experiencing food insecurity in the expected direction. Hypothesis 2 was partially supported, with one measure of social vulnerability significantly, negatively associated with experiencing food insecurity. While the economic dimension of vulnerability is well established in food insecurity research, social, cultural, and political vulnerability should be further explored and refined. However, social exclusion shows promise as a theoretical framework to understand the underlying vulnerabilities and wide range of factors associated with food insecurity.

Previous research on social networks [33,34,50] and social support [32,37] among individuals experiencing food insecurity show that different relationships and different types of support affect food acquisition and coping behaviors. In many cases, food insecure individuals may rely on their family, friends, and acquaintances to exchange or pool resources [33,50,51] or share information about available food resources [50]. In this study, perceptions of perceived positive social support from a spouse, children, and friends were not significantly associated with an increased risk of experiencing food insecurity but perceived positive social support from other family was. Because food insecurity conceptually affects individuals in the same household equally, spouses may not be able to change the risk of experiencing food insecurity, regardless of the level of support between the two partners. While it was expected that individuals with strong, positive relationships with friends could rely on them for support, previous research indicates that individuals who seek food assistance in their social network rely on their family before friends and other acquaintances [50]. Older adults may be unwilling to ask or be uncomfortable asking for aid from their children or friends, so these relationships may not commonly function as a coping mechanism for food insecurity among older adults. In this study, support from other family may have had a significant relationship with food insecurity because other family may be more likely to offer support or have their offer of support accepted, reducing the likelihood of experiencing food insecurity.

It may also be important to examine other social network metrics and types of social support when using the social exclusion framework. For example, the size of the social network may be more important than the quality of the relationships in terms of coping with food insecurity. While the social exclusion framework defined the social dimension as “proximal relationships of support and solidarity” [52], distal relationships may also be important in coping with or preventing food insecurity. For example, larger networks are associated with more access to social resources and may have more information about available food resources which could be shared [50], while a smaller network may not know of as many of the available resources. Furthermore, different types of support should be considered when applying social exclusion to food insecurity. While perceived support has been shown to be associated with food insecurity [35,36], informational and instrumental received support [53] may be a better indicator of social support for this issue. For example, providing transportation assistance [54] and providing direct food assistance [33,37,51] have been shown to be an important type of social support in food acquisition among older adults.

Perceived everyday discrimination was significantly associated with experiencing food insecurity, even after controlling for sociodemographic indicators such as race, gender, and age. Previous research has clearly shown that discrimination and the associated stigma can significantly impact households’ and communities’ material situations. This can be observed in the higher rates of experiencing food insecurity in groups that have been socially or economically marginalized [3]. Whether that marginalization occurs through access to well-paying jobs [55,56], access to housing in communities with resources [55,57,58], the ability to accumulate wealth [56,59], or being denied access to other resources, experiencing discrimination clearly contributes to the likelihood of experiencing food insecurity. Furthermore, there is a significant stigma associated with food assistance programs, including SNAP [60] and using a food pantry or bank [61]. Working to reduce stigma and discrimination overall and in government and non-profit food assistance should be a priority. Furthermore, equity frameworks and anti-racist actions should be built into all nutrition and food insecurity research and interventions.

As expected, as assets and income increased, the risk of experiencing food insecurity decreased. This is well supported by previous research showing the link between economic status [3,62], medical expenses [30,31], and other living expenses [29] with food insecurity. This also theoretically aligns with food insecurity being closely linked to the material resources available to a household, with resources being a key construct in the social exclusion framework [20]. For every USD 10,000 of income, the odds of experiencing food insecurity dropped by 15% in the sample with respondents with any income and 20% in the sample with respondents with incomes less than 400% FPL. There are multiple assistance programs that can provide supplemental income to older adults and which are associated with reduced risk of food insecurity, the major example being SNAP [25], and these programs should be promoted to older adults.

In this study, we used U.S. citizenship status as an indicator for political vulnerability, and it was not significantly associated with experiencing food insecurity. While previous research has shown that older adults who are ineligible for SNAP due to their immigration status are more likely to experience food insecurity [26], the relationship between citizenship status and food insecurity is extremely complex. Citizens within the United States have widely different experiences in terms of the rights and resources they are afforded [63]. Furthermore, non-citizens also have differential access to resources depending on a number of factors, such as their legal status and country of origin. More research is needed to conceptualize and measure political vulnerability. The original definition of political vulnerability in the social exclusion framework is much broader and includes other indicators such as access to resources, including healthcare and housing. When conducting empirical research, the definition of political vulnerability may need to be further refined to be practically measured.

Providing additional economic support to older adults should be a major component of programs and interventions aimed to address food insecurity in older adults, as there is a clear relationship between economic status and the risk of experiencing food insecurity. There are a number of policies that readily could be implemented to increase older adults’ average income (e.g., increasing Social Security Income payments, increasing SNAP benefits, or providing tax credits). Directly providing food through the emergency food system is another way to economically support older adults experiencing food insecurity. Despite the availability of these resources, issues with the utilization of government programs and emergency food resources may be a concern in this population [64,65,66]. That is why efforts to address food insecurity in older adults should aim to reduce, prevent, and address stigma and discrimination associated with using food assistance programs and emergency food services. Finally, including a social component in programs and interventions addressing food insecurity in older adults may increase their effectiveness and provide additional benefits. Explicitly including a social component has previously been shown to be effective in the Older Americans Act nutrition programs [65] and this concept should be more widely implemented.

### 4.1. Limitations

Using secondary data is a major limitation of this analysis. The variables we used were not collected to measure the specific constructs as described in the social exclusion framework. For example, we used U.S. citizenship status as an indicator for political vulnerability, restricting the definition of political vulnerability to the formal recognition of rights. Developing measures to better capture this construct and conducting a primary data collection, while infeasible in this case, would allow researchers to better collect data that truly reflect the political vulnerability of an individual. Despite this, since HRS collects large amounts of information through multiple data collections and questionnaires, we were able to identify indicators that strongly align with the economic, social, and cultural dimensions of vulnerability. Additionally, the survey responses may be biased in several ways, including from surveys collected through different modes, self-selection bias, recall bias, and social desirability bias.

### 4.2. Public Health Implications

These results have important implications for public health policy and food assistance programs. Notably, that food insecurity is not simply an economic issue among older adults and there are other important factors, such as experiencing discrimination or lacking social connections, that should be addressed alongside providing material assistance. As discussed above, there are specific programmatic and policy actions that could be taken in order to prevent older adults from experiencing food insecurity (e.g., increasing benefit amounts, including social components to food assistance interventions, implementing equity frameworks within nutrition and food insecurity prevention programs). Especially given low participation rates, making these programs more appealing to older adults should be a priority, especially for those at increased risk of experiencing food insecurity. As the proportion of older adults as a total of the U.S. population increases, it will be increasingly important to prevent food insecurity in this population in order to improve dietary intake, reduce chronic disease burden, ultimately lower individual and overall healthcare costs, and facilitate longer independent living.

## 5. Conclusions

This study assessed the relationship between indicators of political, economic, social, and cultural vulnerability and experiencing food insecurity. Capturing the larger context described under the social exclusion framework which can lead to the experience of food insecurity will be important for advancing our understanding of this complex issue and developing strategies to address it. While the economic dimension of food insecurity is well studied, further research is needed into the other dimensions of vulnerability contributing to social exclusion and food insecurity among older adults.

## Figures and Tables

**Figure 1 nutrients-13-03896-f001:**
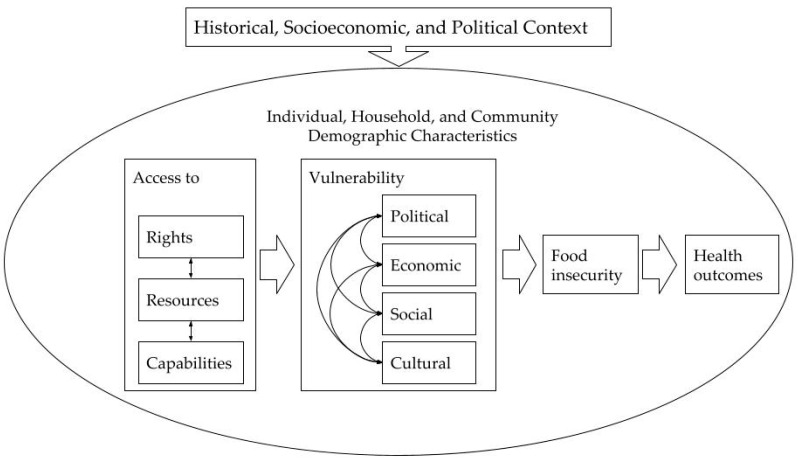
Conceptual model of social exclusion impacting food insecurity and health outcomes through political, economic, social, and cultural vulnerability.

**Figure 2 nutrients-13-03896-f002:**
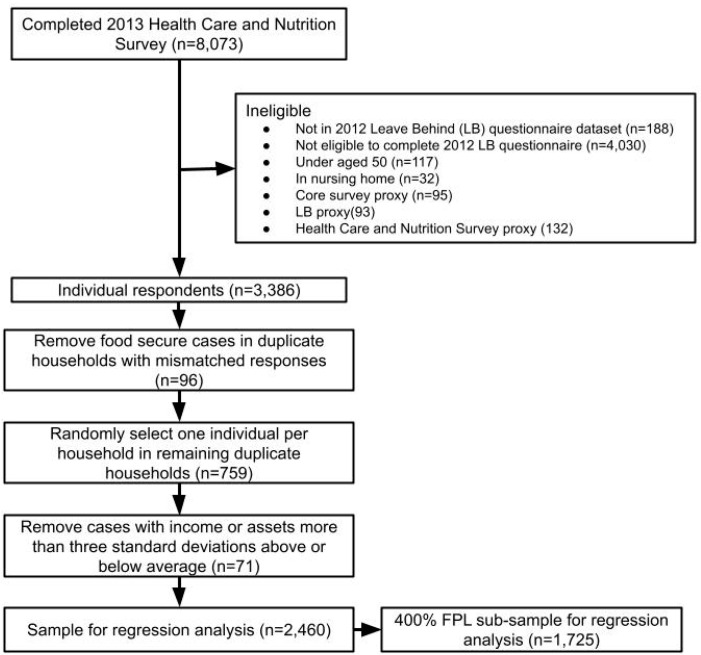
Flow chart showing procedure used to generate the study samples for analysis of the impact of political, economic, social, and cultural vulnerability on food insecurity using Health and Retirement Study data.

**Table 1 nutrients-13-03896-t001:** Descriptive statistics for categorical outcome, associated, and control variables from the Health and Retirement Study used in the regression analysis for individuals with any income (*n* = 2460) and individuals with incomes less than 400% FPL (*n* = 1725).

Variable *Response Options*	All Incomes (*n* = 2460)		Incomes Less than 400% FPL ^1^ (*n* = 1725)	
	*n*	Percent	*n*	Percent
Experienced food insecurity				
* No*	1820	73.98	1153	66.84
* Yes*	640	26.02	572	33.16
U.S. citizenship status				
* U.S. citizen*	2164	87.97	1491	86.43
* Not a U.S. citizen*	296	12.03	234	13.57
SNAP				
* Received SNAP*	290	11.79	269	15.59
* Did not receive SNAP*	2147	87.28	1438	83.36
* Missing*	23	0.93	18	1.04
Health insurance				
* Has health insurance*	2116	86.02	1148	83.94
* Does not have health insurance*	279	11.34	218	12.64
* Missing*	65	2.64	59	3.42
Race				
* White*	1792	72.85	1185	68.70
* Non-White*	659	26.78	532	30.84
* Missing*	9	0.37	8	0.46
Hispanic ethnicity				
* Not Hispanic*	2164	87.97	1470	85.22
* Hispanic*	294	11.95	254	14.72
* Missing*	2	0.08	1	0.06
Gender				
* Male*	913	37.11	561	32.52
* Female*	1547	62.89	1164	67.48
Education				
* Less than high school*	387	15.73	361	20.93
* High school or equivalent*	889	36.14	706	40.92
* Some college*	652	26.50	413	23.94
* College and above*	532	21.63	246	14.20
Marital status				
* Separated/divorced/widowed/never married*	1171	47.60	1013	58.72
* Married/partnered*	1288	52.36	711	41.22
* Missing*	1	0.04	1	0.06
Living Arrangement				
* Living alone*	694	28.21	595	34.49
* Not living alone*	1765	71.75	1130	65.51
*Missing*	1	0.04	0	0.00

^1^ Federal Poverty Level.

**Table 2 nutrients-13-03896-t002:** Descriptive statistics for continuous independent and control variables from Health and Retirement Study used in the regression analysis for individuals with any income (*n* = 2460) and individuals with incomes less than 400% FPL (*n* = 1725).

Variable	All Incomes (*n* = 2460)					Incomes Less than 400% FPL ^1^ (*n* = 1725)				
	Mean	SE ^2^	Min	Max	Missing	Mean	SE ^2^	Min	Max	Missing
Perceived positive social support from spouse	1.34	0.03	0	3	398	1.06	0.03	0	3	294
Perceived positive social support from children	2.00	0.02	0	3	366	2.02	0.03	0	3	262
Perceived positive social support from other family	1.81	0.02	0	3	373	1.82	0.03	0	3	269
Perceived positive social support from friends	1.92	0.02	0	3	372	1.89	0.02	0	3	266
Perceived everyday discrimination	0.56	0.02	0	5	378	0.57	0.02	0	5	269
Assets (in 10,000 US dollars)	18.15	0.58	–168.50	179.80	0	11.54	0.44	–89.99	107	0
Income (in 10,000 US dollars)	5.15	0.10	0	30.29	0	2.65	0.04	0	6.03	0
Medical expenditures (in 10,000 US dollars)	0.29	0.01	0	20.73	0	0.28	0.02	0	20.73	0
Self-rated health	2.93	0.02	1	5	17	3.11	0.02	1	5	13
Activities of daily living	0.27	0.02	0	5	3	0.35	0.02	0	5	3
CESD depression score	1.49	0.04	0	8	1	1.74	0.05	0	8	1
Chronic Conditions	2.18	0.03	0	8	0	2.35	0.04	0	8	0
Age	67.19	0.21	50	100	0	68.30	0.25	50	100	0

^1^ Federal Poverty Level. ^2^ Standard Error.

**Table 3 nutrients-13-03896-t003:** Results of logistic regression estimating the odds of experiencing food insecurity in adults aged 50 years or more as a function of social vulnerability (perceived positive social support from spouses, children, other family, and friends), cultural vulnerability (perceived everyday discrimination), economic vulnerability (value of vehicles, primary residence, and non-housing financial weather and yearly income), and political vulnerability (U.S. citizenship status) and controlling for demographic and health-related factors for individuals with any income (*n* = 2460) and individuals with incomes less than 400% FPL (*n* = 1725).

Variable *Response Options*	All Incomes (*n* = 2460)			Incomes Less than 400% FPL ^1^ (*n* = 1725)		
	Odds Ratio	SE ^2^	*p*-Value	Odds Ratio	SE ^2^	*p*-Value
Perceived positive social support from spouse	0.96	0.01	0.649	0.97	0.11	0.761
Perceived positive social support from children	1.02	0.07	0.739	1.08	0.08	0.345
Perceived positive social support from other family	**0.86**	**0.06**	**0.015**	**0.84**	**0.06**	**0.010**
Perceived positive social support from friends	1.02	0.07	0.836	1.02	0.08	0.835
Living Arrangement (reference = *Not living alone*)						
* Living alone*	1.16	0.18	0.479	1.16	0.20	0.416
Perceived everyday discrimination	**1.68**	**0.15**	**<0.001**	**1.82**	**0.18**	**<0.001**
Assets (in 10,000 US dollars)	**0.97**	**0.01**	**<0.001**	**0.97**	**0.01**	**<0.001**
Income (in 10,000 US dollars)	**0.85**	**0.02**	**<0.001**	**0.80**	**0.04**	**<0.001**
U.S. Citizenship status (reference = *U.S. Citizen*)						
* Not a U.S. Citizen*	1.26	0.25	0.304	1.41	0.31	0.180
SNAP (reference = *Did not receive SNAP*)						
* Received SNAP*	**1.70**	**0.28**	**0.013**	**1.62**	**0.29**	**0.031**
Health insurance (reference = *Has health insurance*)						
* Does not have health insurance*	**1.79**	**0.31**	**0.012**	1.65	0.33	0.052
Household medical expenditures	1.23	0.15	0.134	1.10	0.12	0.404
Self-rated health	**1.16**	**0.08**	**0.049**	1.09	0.09	0.251
Activities of daily living	1.10	0.08	0.223	1.08	0.08	0.314
CESD depression score	**1.09**	**0.03**	**0.005**	**1.09**	**0.04**	**0.011**
Chronic Conditions	**1.14**	**0.05**	**0.005**	**1.19**	**0.06**	**0.001**
Race (reference = *White*)						
* Non-White*	**1.40**	**0.18**	**0.025**	1.32	0.18	0.085
Hispanic ethnicity (reference = *Not Hispanic*)						
* Hispanic*	1.43	0.28	0.127	1.30	0.27	0.276
Gender (reference = *Male*)						
* Female*	1.23	0.16	0.140	1.23	0.17	0.192
Age	**0.95**	**0.01**	**<0.001**	**0.95**	**0.01**	**<0.001**
Education (reference = college and above)						
* Less than High school*	**2.21**	**0.49**	**0.013**	**2.17**	**0.54**	**0.030**
* High school or equivalent*	1.49	0.29	0.089	1.47	0.33	0.157
* Some college or greater*	1.59	0.32	0.059	1.68	0.40	0.088
Marital status (reference = Married/partnered)						
* Separated/divorced/widowed/never married*	0.66	0.18	0.065	0.66	0.20	0.084

Note: Bolded entries indicate a significant value at alpha = 0.05, ^1^ Federal Poverty Level, ^2^ Standard Error.

## Data Availability

All data used can be found at https://hrs.isr.umich.edu/about (accessed on 4 October 2021).

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
