# Peer review of "The Relationship between Political, Economic, Social, and Cultural Vulnerability and Food Insecurity among Adults Aged 50 Years and Older"

_nutrients, 2021, doi:10.3390/nu13113896_

Round 1
Reviewer 1 Report
This is a very interesting study and filling a notable gap in the literature. The authors do an excellent job including various covariates that are known to be associated with food insecurity. However, there are some areas that need to be addressed/modified prior to publication. Please see the comments below.
- Provide domestic prevalence, prevalence among your specific population, and other relevant background information regarding food insecurity that is the same timeframe as when data for this study was collected.
- Please consider putting a conceptual model/framework or some type of figure in the introduction to orient the reader about the various moving parts of this literature base and analysis. You touch on many important gaps in the literature, but at times it feels a bit choppy. I think a figure would be helpful to tie it all together and make it clearer to the reader what gap you are filling (or rather, various notable gaps in the literature!) with this work.
- Given the size of your sample, why was food insecurity dichotomized?
- Given that this is a cross-sectional analysis and thus, cannot fully determine causality, please consider using the terms correlates, factors, or associated instead of predictors/predicting respectively.
- Overall, more description of the various measures is needed (lines 135-159).
- The methods does not make it super clear that your results are stratified by income, please introduce that into the methods.
- Given the number of variables and covariates included in the logistic regression models, did you do any examination of potential collinearity? Please discuss that in the methods, results, and discussion.
- Your discussion could benefit from a section on the public health implications of your findings. I think you have allude to the implications in various areas of the discussion. However, I think that your findings are notable and should be further emphasized and the public health policy and program implications discussed explicitly in it's own section (maybe after Limitations)
Again, this is a very interesting paper with valuable findings. By addressing these areas I think your manuscript will be an even greater contribution to the literature.
Reviewer 2 Report
paper on the determinants of food insecurity. For its high-quality, I only offer some suggestions of improvement and commend the author's for their work.
Introduction:
Line 37. Food insecurity is a substantial public health problem affecting approximately one in ten adults aged 50 years and older. Where? In the world? In high-income countries? In the US?
Line 77-78. Please describe how vulnerability is defined in the social exclusion framework.
Line 87. The hypothesis will look more complete if the direction of the expected significant relation is described.
Results:
I would recommend the use of italics in table to to differentiate the variable names and response options to make it more readable.
Round 2
Reviewer 1 Report
Excellent job addressing mine and the other reviewers comments. This is an incredibly strong manuscript and I look forward to citing it in the future. Thank you for addressing and/or responding to my comments, all potential issues have been resolved and the manuscript is ready for publication.